# Maternal Epigenetic Dysregulation as a Possible Risk Factor for Neurodevelopmental Disorders

**DOI:** 10.3390/genes14030585

**Published:** 2023-02-25

**Authors:** Carla Lintas, Ilaria Cassano, Alessia Azzarà, Maria Grazia Stigliano, Chiara Gregorj, Roberto Sacco, Andrea Stoccoro, Fabio Coppedè, Fiorella Gurrieri

**Affiliations:** 1Research Unit of Medical Genetics, Department of Medicine, Università Campus Bio-Medico di Roma, Via Alvaro del Portillo, 21, 00128 Roma, Italy; 2Operative Research Unit of Medical Genetics, Fondazione Policlinico Universitario Campus Bio-Medico, Via Alvaro del Portillo, 200, 00128 Roma, Italy; 3Operative Research Unit of Hematology, Stem Cell Transplantation, Transfusion Medicine and Cellular Therapy, Fondazione Policlinico Universitario Campus Bio-Medico, Via Alvaro del Portillo, 200, 00128 Roma, Italy; 4Medical Genetics Laboratory, Department of Translational Research and of New Surgical and Medical Technologies, University of Pisa, 56126 Pisa, Italy

**Keywords:** neurodevelopmental disorders, epigenetic dysregulation, exposome, DNA methylation, risk factors, Autism Spectrum Disorder, Attention Deficit Hyperactivity Disorder, Asperger Syndrome

## Abstract

Neurodevelopmental Disorders (NDs) are a heterogeneous group of disorders and are considered multifactorial diseases with both genetic and environmental components. Epigenetic dysregulation driven by adverse environmental factors has recently been documented in neurodevelopmental disorders as the possible etiological agent for their onset. However, most studies have focused on the epigenomes of the probands rather than on a possible epigenetic dysregulation arising in their mothers and influencing neurodevelopment during pregnancy. The aim of this research was to analyze the methylation profile of four well-known genes involved in neurodevelopment (*BDNF*, *RELN*, *MTHFR* and *HTR1A*) in the mothers of forty-five age-matched AS (Asperger Syndrome), ADHD (Attention Deficit Hyperactivity Disorder) and typically developing children. We found a significant increase of methylation at the promoter of the *RELN* and *HTR1A* genes in AS mothers compared to ADHD and healthy control mothers. For the *MTHFR* gene, promoter methylation was significantly higher in AS mothers compared to healthy control mothers only. The observed dysregulation in AS mothers could potentially contribute to the affected condition in their children deserving further investigation.

## 1. Introduction

Neurodevelopmental Disorders (NDs) are a heterogeneous group of disorders [1] including Autism Spectrum Disorders (ASD), Attention Deficit Hyperactivity Disorders (ADHD), Communication Disorders, Intellectual Disability (ID), Movement Disorders and Specific Learning Disorders (SLD). They are characterized by biological neurodevelopmental alterations occurring during the first trimester of pregnancy and consist of anomalies of brain maturation during the prenatal and postnatal periods. Within ASD, Asperger Syndrome (AS) is considered the least severe form of autism and individuals affected by AS are often referred as high-functioning individuals.

NDs are considered multifactorial diseases with both genetic and environmental components [2,3]. The contribution of each component varies among cases: in some individuals the genetic component is prevalent, in others the environmental component prevails, whereas in some others, the contribution of the two components weigh approximately the same.

The genetic contribution is highly complex as it is based on the interaction of hundreds of loci including both common and rare CNVs (Copy Number Variants) and SNVs (Single Nucleotide Variants) [4,5]. The genetic underpinnings of NDs have unveiled an unpredicted degree of heterogeneity and complexity, ranging from rare variants endowed with full penetrance to common variants each explaining very small proportions of the overall phenotypic variance either alone or through gene x environment interactions [4,5]. From a practical point of view, the molecular diagnosis of NDs, mainly for ASD, is a multistep process involving different types of pangenomic analysis: array-CGH is the first-tier quantitative genetic test in many countries, followed by qualitative tests such as whole or targeted exome sequencing.

Environmental factors contributing to NDs in the offspring include smoking and alcohol consumption, delivery complications, stress, diet, toxic agent exposure and viral injury during pregnancy and early postnatal life [6,7,8]. Environmental factors influence the individual’s epigenetic status by acting on molecular mechanisms such as DNA methylation and/or histone modifications. Indeed, DNA methylation and gene transcription profiles represent the so-called “internal exposome”; conversely, the individual response to whatever is in the outside environment is the “external exposome”. Based on this principle, levels of DNA methylation/gene expression in different health conditions can be considered a biomarker of exposure, resilience and disease risk. Indeed, DNA methylation of CpG dinucelotides frequently occurs nearby the promoter of many genes and results in gene transcription repression, whereas unmethylated gene promoters are correlated with gene expression. Additionally, histone modifications such as acetylation, phosphorylation and methylation at specific amino acid residues on their tails, and chromatin remodeling proteins, contribute to controlling gene expression by changing the status of chromatin from an inactive conformation (heterochromatin) to an active conformation (euchromatin).

Environmentally driven epigenetic modifications in genes involved in neurotransmission and brain plasticity may result in altered gene expression leading to maternal distress. Such a combination of factors can then be transmitted to the developing fetus via the placenta or via the maternal HPA (Hypothalamic Pituitary Adrenal) axis [9].

The serotonergic system influences many aspects of mammalian physiology, including cognition, mood, learning and memory, and changes in the expression or function of serotonin receptors have been shown to be correlated with the genetic background [10]. Indeed, deficits in the serotonin-1A receptor encoded by the *HTR1A* gene have been associated with depression and anxiety in heterozygous knockout mice and in their offspring [11]. Serotonin is crucial for fetal brain development and its transplacental passage from the mother to the fetus has been found to be altered in mothers affected by depression during pregnancy [12].

The *RELN* gene was initially studied in the context of neurodevelopment, but it also has a role in the adult brain, particularly in synaptic strength and in higher cognitive functions [13,14]. In addition, reelin exerts proteolytic activity on extracellular matrix proteins which is inhibited by pesticides such as organophosphates [15].

The brain-derived neurotrophic factor (BDNF), encoded by the *BDNF* gene, also plays an important role during pregnancy by influencing brain development in the fetus [16]. BDNF and other neurotrophins are involved in embryo implantation, placental development and maturation, vascular growth modulation and maternal immunity. Initially they are transported from the mother across the utero-placental barrier, but after the second trimester, they are also produced by the placenta during fetal growth and development. Preeclampsia, severely elevated blood pressure during pregnancy, has been associated with lower maternal BDNF concentrations [17] and with higher incidence of NDs in the offspring [18], including ASD and ADHD. Maternal nutrition as well as genetic background affect and influence directly or indirectly, through epigenetics, the levels and expression of neurotrophins [16].

Among dietary factors, folate supply during the periconceptional period and the first trimester of pregnancy [19,20] is considered one of the most important environmental factors influencing correct neurodevelopment. Indeed, its deficiency has been associated with developmental anomalies such as spina bifida and anencephaly [21], but also with less severe conditions including intellectual disability, ADHD [22] and ASD [23]. From a biological point of view, folate is the main methyl source converting homocysteine into methionine, which in turn is a precursor of *S*-adenosylmethionine (SAM), the universal methyl donor for DNA and histones, as well as for RNA, phospholipids and proteins [24,25]. Folate and folate-related pathways are regulated by many different enzymes encoded by genes whose polymorphisms have been associated with many different types of conditions and diseases [26,27]. Therefore, besides folate supply during pregnancy, genetic and epigenetic variants in folate-related pathway genes may also be considered risk factors for NDs [27]. Indeed, dynamic changes in DNA methylation and biosynthesis occur following egg fertilization and continue during embryo development [28,29,30]. The most extensively studied gene of folate metabolism is the *MTHFR* gene encoding for methylenetetrahydrofolate reductase, an enzyme that catalyzes the conversion of 5,10-methylenetetrahydrofolate to 5′-methyltetrahydrofolate, which is essential for homocysteine conversion to methionine. The maternal to fetal transfer of folate is mediated by the placental folate receptor which preferentially binds to 5′-methyltetrahydrofolate. Hence variations in folate isoform concentration due to maternal *MTHFR* genetic and/or epigenetic variants could affect DNA methylation and nucleotide synthesis downstream in the developing embryo.

In the light of these considerations, we hypothesize that mothers of children affected by AS or ADHD may carry epigenetic alterations compared to age matched mothers of healthy children. Such alterations may increase the risk for AS or ADHD in their children. These epigenetic changes may be induced by environmental factors, for instance pollution, viral infections, stress and the diet during the mother’s life span or also by the “psychological mental status” of the mother herself. In order to test our hypothesis, we assessed promoter methylation in four well-known genes playing a role in neurodevelopment: *BDNF*, *MTHFR*, *RELN* and *HTR1A*. These four genes were also selected because their expression is controlled by promoter DNA methylation as suggested by the presence of CpG islands near their promoter regions.

## 2. Materials and Methods

### 2.1. The Experimental Sample

A total of 135 subjects were recruited at Fondazione Campus Bio-medico of Rome (Italy). The subjects included 45 women who had at least one child affected by Attention Deficit Hyperactivity Disorder (ADHD mothers), 45 women who had at least one child affected by Asperger Syndrome (AS mothers) and 45 women with healthy children (HC mothers). Healthy control mothers (HC mothers) were included only if they had at least one full-term pregnancy and no children affected by neuropsychiatric/neurodevelopmental disorders or other disorders. The mothers’ age at sampling ranged from 33 to 55 years and was not significantly different among the three groups. ADHD diagnosis was assessed with standard psychodiagnostic tests including Griffith Mental Developmental Scales (GMDS-ER) [31], Child Behaviour Check List (CBCL) [32] for parents and for teachers and Conner’s Rating Scales (CRS-R) [33]. ASD diagnosis was assessed with Autism Diagnostic Observation Schedule (ADOS) [34] and Autism Diagnostic Interview—Revised (ADI-R) [35]. All subjects had the same ethnicity and geographic origin. For all subjects, exclusion criteria included (1) smoking, (2) current use of drugs knowing to interfere with DNA methylation and (3) current use of folic acid or vitamin B supplements. A questionnaire mainly related to the environmental risk factors of the mother during pregnancy and her life span was administered to the mothers (Appendix A). All participants to the study signed a consent form. The study was also approved by the Ethics Committee of the University Campus Bio-medico of Rome (see specific section at the end of the manuscript).

### 2.2. Sample Collection and DNA Isolation

Peripheral blood was collected in two EDTA tubes and genomic DNA was extracted using the QIAamp DNA Blood Mini Kit (Qiagen, Milan, Italy) following the manufacturer’s instructions. In order to reduce bias, samples relative to the three different groups (ADHD, AS and controls) were treated simultaneously.

### 2.3. Bisulfite Treatment and High-Resolution Melting Analysis

About 750 ng of genomic DNA was subjected to bisulfite conversion using the Epitect Bisulfite Kit (Qiagen, Milan, Italy) following the manufacturer’s instructions. Sodium bisulfite converts all unmethylated cytosine into uracil, and methylated cytosine is left unchanged. Primer sequences for the four tested genes and PCR amplification conditions are reported in Table 1. Primers were already used successfully in previous studies. In particular, primers for the *MTHFR* gene were derived from [36], those for *BDNF* from [37], and those for the *RELN* and *HTR1A* genes from [38]. The Methylation-Sensitive High-Resolution Melting (MS-HRM) method was used to quantify the mean level of methylation of each sample. Overall, 20 ng of converted DNA was used, in addition to 1X of EpiTect HRM PCR mix (Qiagen, Milan, Italy), ultrapure water and 1 picomole of each primer in a final volume of 20 μL. PCR amplification was carried out using the 7900 HT Fast Real-Time PCR System (Thermo Fisher Scientific, Monza, Italy) according to the following protocol: one step of denaturation at 95 °C for five minutes, 40 cycles at 95 °C for 30 s, annealing temperature (Table 1) for 30 s, 72 °C for 30 s. The plate was immediately centrifuged and subject to an HRM step which consisted of a single step of 95 °C for 15 s followed by temperature from 50 °C to 95 °C using a ramp rate of 1% withholding steps of 60 s and 15 s, respectively; a final step at 60 °C for 15 s was performed. Each assay included a negative control, unconverted genomic DNA and seven methylation standards (EpiTect PCR control DNA set, Qiagen, Milan, Italy) prepared by mixing fully methylated and unmethylated converted DNA in known proportions (0, 10, 20, 30, 40, 50 and 100% methylated DNA). Each sample was run in triplicate, whereas methylation standards were run in duplicate. Temperature and fluorescence data relative to the melt curve of each sample and standard were exported from the SDS 2.4 software (Thermo Fisher Scientific, Monza, Italy) and used to derive interpolation curves. The interpolation method described elsewhere [39] was used to obtain a single mean methylation value for each sample rather than a methylation range.

### 2.4. Statistical Analysis

Parametric ANOVA followed by multiple comparisons test (LSD and Tamhane), Kruskal–Wallis non-parametric ANOVA and Mann–Whitney U test were employed to compare the gene methylation level of each of the four genes among the three groups (ADHD mothers, AS mothers and healthy control mothers). Correlation analyses between methylation level and clinical variables were performed using non-parametric Kendall’s τ statistics; gene methylation levels are reported as mean ± standard error of the mean (S.E.M.). Statistical significance is set at a nominal two-tail *p* < *0*.05, unless otherwise specified. Statistical analyses were performed using SPSS software release 27.0 (SPSS INC, Chicago, IL, USA) and R-4.2.2 software.

## 3. Results

### 3.1. AS Mothers Display a Higher Methylation Level Than ADHD and Healthy Control Mothers

The mean percentage of methylation among the three groups was significantly different for the *HTR1A* and the *RELN* genes (F = 14.080, *p* = 0, *n* = 2; F = 8.240, *p* = 0, *n* = 2, respectively), nearly significant for the *MTHFR* gene (F = 2.978, *p* = 0.054, *n* = 2) and not significant for the *BDNF* gene (Table 2 and Figure 1). In general, the mean percentage of methylation level was the highest in the AS group for all four genes. The difference between AS mothers and ADHD or control mothers was statistically significant for *HTR1A* gene (AS vs. ADHD: LSD = 1.92, *p* = 0, AS vs. control: LSD = 2.26, *p* = 0) and for the *RELN* gene (AS vs. ADHD: LSD = 3.09, *p* = 0.001, AS vs. control: LSD = 3.38, *p* = 0). By contrast, differences in methylation percentage between ADHD and control mothers were not significant for these two genes (ADHD vs. control: LSD = 0.34, *p* = 0.460 for *HTR1A*, ADHD vs. control: LSD = 0.29, *p* = 0.750 for *RELN*) and also for *BDNF* and *MTHFR* genes. For the *MTHFR* gene, the mean percentage of methylation was significantly different for AS mothers versus control mothers (LSD = 2.66, *p* = 0.020), but not for AS mothers versus ADHD mothers (LSD = 1.94, *p* = 0.087). With the exception of the *BDNF* gene, the percentage of methylation was always very similar in ADHD mothers and healthy control mothers and was always slightly higher in the first group compared to the second one (Table 2 and Figure 1). Indeed, for the *BDNF* gene, methylation levels of control mothers were more similar to those relative to AS mothers (Table 2 and Figure 1). Among the four tested genes, *MTHFR* had the highest mean overall percentage of methylation at 33.8 ± 5.4, whereas the other three genes were nearly all unmethylated (3.5 ± 4.6 for *BDNF*, 3.6 ± 2.4 for *HTR1A*, 3.5 ± 4.6 for *RELN*).

### 3.2. Mean Percentage of Methylation Correlates with Some Clinical Variables

In ADHD mothers, the *BDNF* gene’s percentage of methylation correlates negatively with newborn birth weight (tau di Kendall = −0.387 and *p* = 0.008) and positively with threatened miscarriage (tau di Kendall = 0.409 and *p* = 0.017) and infections (tau di Kendall = 0.356 and *p* = 0.038) during pregnancy (Table 3). For the other two groups no significant correlations between *BDNF* percentage of methylation and clinical variables were found.

With regard to the *HTR1A* gene, in AS mothers the methylation percentage was negatively correlated with newborn birth weight (tau di Kendall = −0.274 and *p* = 0.013) and positively with threatened miscarriage (tau di Kendall = 0.505 and *p* = 0.025) during pregnancy (Table 3). In healthy control mothers, a significant negative correlation was detected between *HTR1A* methylation percentage and obstetric complications in the newborn (tau di Kendall = −0.327 and *p* = 0.028).

No significant correlations were found for the *RELN* and *MTHFR* genes in all the three groups (Table 3).

## 4. Discussion

Many studies have documented a higher risk for NDs in the offspring of mothers exposed to different environmental hazards during pregnancy. Maternal exposure to the anti-epileptic drug Valproic acid (VPA) during pregnancy increases the risk for ASD and altered expression of CNS markers such as reelin or NGF, and deficient social interactions have been reported in their offspring [40]. Another study has documented how maternal thyroid hormone deficiency affects neocortical cytoarchitecture and influences reelin expression in the offspring [41,42]. The role of maternal viral or bacterial infection in the first trimester of pregnancy in increasing the incidence of ASD has been largely documented in the literature [43]. Maternal mental distress [9] and malnutrition during pregnancy [16] have also been correlated with altered neurodevelopment in the offspring. However, nearly all these studies have focused on finding a causal link between an environmental adverse factor during pregnancy and the offspring phenotype. However, very few studies to date have concentrated on understanding how the biology or the “health status” of the pregnant women may change in response to exposure to adverse conditions during pregnancy and/or the periconceptional period. Our study documents for the first time that the methylation profile of a panel of genes involved in neurodevelopment is significantly different in the mothers of children affected by AS compared to mothers of children affected by ADHD and mothers of typically developing children. Methylation levels of *RELN* and *HTR1A* were significantly higher in AS mothers compared to ADHD and healthy control mothers, suggesting that expression of these two genes may be lower in AS mothers. In ADHD mothers, the methylation level for these two genes was in between AS and healthy control mothers, though it was much closer to the control group. This observation is very interesting and can be explained by the fact that in general ADHD is a less severe neurodevelopmental disorder compared to autism spectrum disorder. These two genes play an important role in neurodevelopment, but also in adult cognitive functions and their observed epigenetic changes in AS mothers. Changes driven by external factors such as diet and chemical exposure, or by internal factors such as psychological maternal stress, could have greatly affected the cell state and could have contributed to AS onset in their offspring. Within this context, it is interesting to note the significant positive correlation observed in AS mothers between *HTR1A* methylation levels and miscarriage event during pregnancy, as well as the negative correlation between *HTR1A* methylation levels and birth weight (Table 3). Altered *HTR1A* or other serotonergic genes’ methylation profiles have been documented in psychiatric diseases such as suicidal behavior [44], panic disorder [45], depression and anxiety disorder [46].

A recent study [11] on 5HT1AR heterozygous knockout (Het) mice demonstrated that 5HT1AR deficit causes sex-biased outcomes in the offspring by altering the maternal immune system and midgestational in utero environment. Female offspring display immunodysregulation whereas male offspring exhibit an anxiety-like behavior.

Alterations in the expression of genes (as the *HTR1A* gene) involved in the serotonergic pathway in the pregnant mothers may thus affect transplacental serotonin passage from the mother to the fetus increasing the risk for AS as suggested by our data.

Abnormal *RELN* gene methylation and consequent downregulation has been linked to many neuropsychiatric and neurodevelopmental disorders including ASD, schizophrenia, bipolar disorder, major depression and Alzheimer’s disease [47]. In a previous study we found increased methylation at the *RELN* gene promoter on ASD post-mortem brains compared to control brains [14]. Sex differences in *RELN* methylation were observed in the blood DNA of ASD individuals. Indeed, *RELN* methylation was higher in males than in females [48]. In ASD females, blood *RELN* methylation levels were inversely correlated with disease severity, as assessed through the ADOS-2 score [38]. In males, blood *RELN* methylation levels were linked to the methylation levels of *EN2*, a gene that protects dopaminergic neurons against mitochondrial insult from pesticides [48]. Reduced RELN in the mother can directly affect neurodevelopment of her offspring, or alternatively the hypermethylation profile can be “transmitted” to the fetus via maternal germline. Indeed, recent research work has demonstrated the existence of epigenetic inheritance across one or more generations in mammals [49,50]. When the change is passed through the germline to the next generation it becomes a heritable epigenetic mark. The latter can last just in one generation (intergenerational epigenetic inheritance) or can be passed on and maintained for many generations (transgenerational epigenetic inheritance). We therefore cannot rule out the hypothesis that the increased *RELN* methylation observed in AS mothers in our study has been passed through the germline to their AS child. We aim to investigate this hypothesis in our future work with the objective of finding family-specific epigenetic signatures shared by the mother and the affected offspring. Indeed, lower reelin concentration linked to increased *RELN* gene methylation in the developing embryo could significantly impact CNS formation. Furthermore, reelin has also been shown to influence the expression of serotonin receptors [51] in the frontal cortex of the reeler mouse as well as in schizophrenic patients. Therefore, we cannot exclude also that the methylation increase at the *RELN* and *HTR1A* genes observed in AS mothers is the result of a direct or indirect interaction between these two genes.

*MTHFR* methylation percentage was significantly higher in AS mothers compared to control mothers, while no significant differences were observed between AS mothers and ADHD mothers or between ADHD mothers and control mothers as previously reported [52]. Since *MTHFR* is the most important gene in regulating the amount of folate supply to the embryo, a hypermethylation of its promoter could result in less folate availability for methylation reactions during neurodevelopment with consequent increase in AS risk [27]. Indeed, recent literature meta-analyses support a role for the *MTHFR* 677C > T polymorphism, the main polymorphism impairing MTHFR activity, as a risk factor for ASD [53,54]. Moreover, increased *MTHFR* promoter methylation results in decreased *MTHFR* gene expression levels and has been associated with preeclampsia [55], recurrent miscarriages [56,57], trisomy 21 and congenital heart defects in the offspring [58,59] as well as with male infertility, likely inducing impaired methylation in sperm cells [60,61]. Collectively, these studies suggest that impaired *MTHFR* methylation in one of the parents might result in transmissible epigenetic changes which, in turn, could negatively impact embryo development and/or pregnancy outcome. In this regard, during the last few years, research work has demonstrated the existence of epigenetic inheritance across one or more generations in mammals [49,50]. An epigenetic change can arise sporadically in an individual or by exposure to some environmental stimulus. When the change is passed through the germline to the next generation it becomes a heritable epigenetic mark. The latter can last just in one generation (intergenerational epigenetic inheritance) or can be passed and maintained for many generations (transgenerational epigenetic inheritance).

This mechanism could also explain why many familial forms of AS often remain without a molecular diagnosis even after the multistep genetic diagnostic process routinely performed (see introduction for more details). Indeed, the existence of an “autism spectrum” in some families in which more than one member is affected by AS (familial forms of AS), could be explained by intergenerational or even transgenerational epigenetic inheritance as the etiological mechanism underpinning the familial forms of Asperger Syndrome.

In summary, this study has shown that AS mothers have significantly higher methylation at the *HTR1A* and *RELN* gene promoters compared to ADHD and control mothers. They also display significantly higher promoter methylation at the *MTHFR* gene promoter compared to control mothers only. These epigenetic changes may be passed from the mother to the fetus directly through the germline, or alternatively, they could impact the neurobiology of the fetus by downregulating the availability of gene products via the placenta and/or via the maternal HPA axis. Future studies are needed to shed light on this aspect by extending methylation analysis to the AS probands and by performing transcriptome studies in the mothers and in their affected children.

## Figures and Tables

**Figure 1 genes-14-00585-f001:**
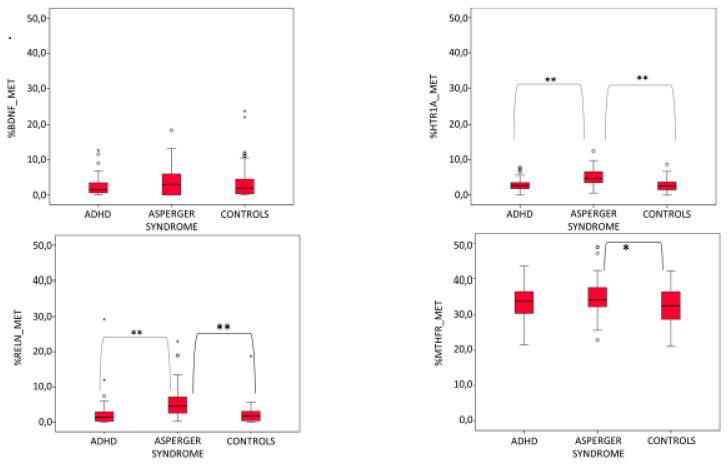
Percentage of methylation for the *BDNF, HTR1A, RELN* and *MTHFR* genes in ADHD, AS and healthy control mothers (** = difference between groups statistically significant, * = difference between groups nearly significant, *p* = 0.054).

**Table 1 genes-14-00585-t001:** PCR conditions, location of CpGs and dinucleotides, and primer sequences for PCR amplification.

Genes	Primer Sequences	Number of CpGs in the Amplified Region	Anealling Temperature (°C)	PCR Product Size (bp)
*MTHFR*	5′-TTTTAATTTTTGTTTGGAGGGTAGT-3′ (for)	7	55	155
	5′-AAAAAAACCACTTATCACCAAATTC-3′ (rev)			
*BDNF*	5′-GGGTTGTTAATTTATATTTGGGAAGT-3′ (for)	4	58	119
	5′-AACCACTAATTACCCACAAAAACC-3′ (rev)			
*HTR1A*	5′-TGTTTGTTAGTGGGGAGATTTTAGT-3′ (for)	15	52	251
	5′-CAAAAACCCAAACAAAAAATTCTTA-3′ (rev)			
*RELN*	5′-TTGAAGAGTTTAGAAGTAATGAATAATAGA-3′ (for)	7	56	192
	5′-ACCTCATCTATAAAAAATTTTAAAATAAAA-3′ (rev)			

**Table 2 genes-14-00585-t002:** ANOVA test results for percentage of methylation of the four studied genes in AS mothers, ADHD mothers and healthy control mothers. df = degree of freedom, F = Fisher value, Sig = statistically significant. In bold significant values.

	Status	N.	Mean ± SD	df	F	Sig.
%BDNF_MET	ADHD	45	2.5 ± 3.0	2	1.608	0.204
	ASD	43	4.1 ± 4.7			
	CTRL	45	3.9 ± 5.6			
	Tot.	133	3.5 ± 4.6			
%HTR1A_MET	ADHD	45	3.0 ± 1.9	2	14.080	**0.000**
	ASD	45	5.0 ± 2.7			
	CTRL	45	2.7 ± 1.8			
	Tot.	135	3.6 ± 2.4			
%RELN_MET	ADHD	45	2.6 ± 4.7	2	8.240	**0.000**
	ASD	44	5.7 ± 5.0			
	CTRL	45	2.3 ± 3.1			
	Tot.	134	3.5 ± 4.6			
%MTHFR_MET	ADHD	45	33.4 ± 4.8	2	2.978	**0.054**
	ASD	44	35.4 ± 5.6			
	CTRL	45	32.7 ± 5.6			
	Tot.	134	33.8 ± 5.4			

**Table 3 genes-14-00585-t003:** Correlations between percentages of methylation for the four genes tested in this study and clinical variables. Statistically significant correlations are highlighted in bold.

			Newborn Birth Weight	Active Smoking	Passive Smoking	Alcohol Use	Drug Use	Threatened Miscarriage	Obstetric Complications- Mother	Obstetric Complications- Newborn	Pregnancy Duration	Exposure to Toxic/Polluting Agents	Family History of Neuropsychiatric Disorders	Autoimmune/ Inflammatory Diseases	Infections	Psychological Trauma
%BDNF_MET	ADHD	Tau	**−0.387**	0.017	0.260	−0.154	0.244	**0.409**	0.205	0.204	0.191	0.007	−0.033	0.035	**0.356**	0.218
		Sig.	**0.008**	0.920	0.128	0.367	0.154	**0.017**	0.232	0.234	0.266	0.967	0.849	0.838	**0.038**	0.203
		N.	25	25	25	25	25	25	25	25	25	25	25	25	25	25
	ASD	Tau	−0.019	0.225	−0.073		0.000	−0.063	−0.507	−0.133	−0.034		0.364	−0.090		0.418
		Sig.	0.873	0.387	0.780		1.000	0.794	0.052	0.390	0.805		0.163	0.550		0.109
		N.	39	12	12	12	12	14	12	31	36	12	12	33	11	12
	CTRL	Tau	−0.109	−0.249	0.147	−0.206	−0.156	−0.031	0.052	−0.217	−0.180	−0.166	−0.091	−0.156	−0.249	−0.142
		Sig.	0.401	0.100	0.332	0.174	0.303	0.837	0.731	0.152	0.230	0.273	0.549	0.303	0.100	0.348
		N.	32	32	32	32	32	32	32	32	32	32	32	32	32	32
%HTR1A_MET	ADHD	Tau	0.000	0.102	−0.025	−0.236	−0.309	0.198	−0.069	−0.087	−0.139	−0.213	0.055	−0.030	0.025	−0.195
		Sig.	1.000	0.548	0.882	0.166	0.069	0.244	0.683	0.610	0.415	0.210	0.744	0.861	0.882	0.252
		N.	25	25	25	25	25	25	25	25	25	25	25	25	25	25
	ASD	Tau	**−0.274**	0.145	−0.372		0.189	**0.505**	−0.097	0.216	−0.153	−0.261	−0.041	0.042	−0.334	0.261
		Sig.	**0.013**	0.554	0.128		0.440	**0.025**	0.693	0.141	0.245	0.285	0.866	0.768	0.192	0.285
		N.	41	13	13	13	13	15	13	33	38	13	13	35	12	13
	CTRL	Tau	−0.073	0.151	0.133	0.040	−0.289	−0.027	0.187	**−0.327**	−0.033	−0.081	0.035	0.178	−0.052	0.058
		Sig.	0.567	0.311	0.372	0.787	0.053	0.855	0.210	**0.028**	0.821	0.586	0.815	0.231	0.726	0.697
		N.	32	32	32	32	32	32	32	32	32	32	32	32	32	32
%RELN_MET	ADHD	Tau	−0.054	−0.224	−0.019	−0.250	−0.156	0.005	0.005	−0.078	0.178	−0.144	−0.033	−0.035	0.019	0.131
		Sig.	0.707	0.192	0.911	0.145	0.363	0977	0.977	0.650	0.298	0.402	0.849	0.838	0.911	0.444
		N.	25	25	25	25	29	25	25	25	25	25	25	25	25	25
	ASD	Tau	0.036	0.097	0.083		0.075	−0.345	0.048	−0.092	−0.187	−0.065	0.207	0.156	−0.111	−0.261
		Sig.	0.744	0.693	0.735		0.758	0.126	0.844	0.531	0.161	0.789	0.398	0.280	0.664	0.285
		N.	40	13	13	13	13	15	13	33	37	13	13	34	12	13
	CTRL	Tau	−0.237	−0.230	−0.048	−0.213	−0.263	−0.035	0.147	−0.073	−0.179	−0.082	0.145	−0.013	−0.230	0.035
		Sig.	0.066	0.128	0.753	0.158	0.081	0.819	0.331	0.627	0.231	0.585	0.335	0.932	0.128	0.815
		N.	32	32	32	32	32	32	32	32	32	32	32	32	32	32
%MTHFR_MET	ADHD	Tau	−0.094	−0.238	0.063	−0.189	0.000	0.248	0.208	−0.173	−0.050	−0.114	0.065	0.050	0.025	−0.119
		Sig.	0.512	0.161	0.711	0.267	1.000	0.145	0.221	0.308	0.767	0.503	0.703	0.771	0.882	0.484
		N.	25	25	25	29	29	25	25	25	25	25	25	25	25	25
	ASD	Tau	0.003	−0.055	−0.143		0.477	0.091	0.000	−0.071	−0.172	−0.112	0.072	0.143	−0.172	−0.187
		Sig.	0.981	0.830	0.578		0.064	0.698	1.000	0.631	0.200	0.663	0.781	0.322	0.526	0.468
		N.	40	12	12	12	12	14	12	32	37	12	12	34	11	12
	CTRL	Tau	−0.063	−0.174	0.137	−0.234	−0.076	−0.010	−0.042	0.051	0.129	0.290	−0.058	0.093	0.151	0.162
		Sig.	0.623	0.243	0.359	0.116	0.608	0.945	0.776	0.732	0.383	0.052	0.697	0.531	0.312	0.276
		N.	32	32	32	32	32	32	32	32	32	32	32	32	32	32

## Data Availability

Research data are not shared.

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
