# Peer review of "Maternal Epigenetic Dysregulation as a Possible Risk Factor for Neurodevelopmental Disorders"

_genes, 2023, doi:10.3390/genes14030585_

Round 1
Reviewer 1 Report
In this manuscript, Lintas et al., aim to determine if there is an association between maternal epigenetic (DNA methylation) signatures at candidate gene loci and the presence of neurodevelopmental disorders (NDDs) in their offspring. The authors identify subtle but statistically significant increases in the methylation of regions proximal to HTR1A, RELN and MTHFR in peripheral blood of the mothers of children diagnosed with Asperger’s syndrome (AS). Given the complex, and often poorly understood, mechanistic basis of NDDs the subject matter of this paper is likely to be interesting to a broad readership. The study also discusses the concept of intergenerational epigenetic inheritance, although methylation has not been investigated in the proband and therefore this study has limited scope to address this directly. The paper is well written, logically constructed and clear. However, the data is somewhat limited and only acceptable given the reasonably large numbers of control and test human samples included. In this reviewer’s opinion the article is of sufficient interest to justify publication if the authors can make the following changes / additions and address the reviewer’s concerns satisfactorily.
Major comments / queries.
Given that the study involves human samples, the methods section needs to include information regarding sample anonymization procedures, consent and ethical approval.
The introduction should provide a brief section on the nature and suggested role of DNA methylation. E.g. Methylation distribution and proposed repressive function. This provides a link to a suggested figure addition to include below.
The locations (relative to the candidate gene), no. of CpGs and size of each amplicon should be presented in the main data figure to orient the findings and their potential mechanistic insight. The authors suggests that the methylation may have a regulatory consequence and inclusion of the location relative to the target genes in particular is important for this.
Query - did you either sequence the amplicons from each individuals or perform melt curve analysis on them prior to bisulfite conversion. Such a check is important to determine if SNPs between your individuals and the control epitect DNA resulted in altered melting properties that could be mis-interpreted as changes in DNA methylation? This is particularly important as methyl CpG are hyper-mutable and therefore disproportionately mutated in heavily methylated vertebrate genomes.
Query - Is the method appropriate for use on amplicons containing multiple CpG sites (see table S2)? The calibrating DNA will be either fully methylated or un-methylated at all CpGs and, where mixed in different ratios, will still contain fully methylated and unmethylated alleles only. In contrast the test subjects will likely have mosaic methylation across the CpGs within an allele and therefore provide distinct melt values from any that are present in the calibrator samples. Is this method not only appropriate for single CpG site amplicons?
Was methylation checked against the age of the mother? It is well known that methylation changes with age and that risk of NDD does also. Were the mothers of AS patients older and does methylation changes at the test sites also change in control mothers if you include maternal age as a test criteria?
Some commentary on the use of peripheral blood (PB) should be included. PB is obviously used as it is available and relatively non-invasive, however it should be noted in the discussion that the significance of these changes in the mother are unclear e.g. does this indicate that all tissues are affected in the mother or that a defect in the blood specifically is responsible for the NDD in the ofspring (e.g. immune response / changed blood cellular composition)?
Minor comments.
L88 – ‘anomalies as spina bifida’ – should read – anomalies such as spina bifida.
L153 – ‘The plate was immediately spinned and’ – should read - ‘The plate was immediately centrifuged and’
Author Response
Dear reviewer,
We thank you very much for having read our manuscript and have provided us with very useful comments.
Please find enclose the file with our answers to your comments. We did our best to include all your suggestions in the manuscript.
Best Regards,
Carla Lintas

Reviewer 2 Report
Excellent study investigating the role of methylation in neurodevelopment genes of mothers and children with AS, ADHA, and developing children. I particulary enjoyed the results pertaining to the HTR1A and RELN genes which were significantly methylated. One point I have with regards to the methylation of BDNF and MTHFR is that perhaps these genes were not signifcantly changed because perhaps the methylation (if present) could have been outside the promoter region and perhaps in the gene body? Whilst on the results section, I think that the authors could improve the clarity of the graphs as it is blurry and the lines denoting signifcance between groups are a bit messy. To make the table of correlations slightly concise, I would recommend only including parameters that were significant and leaving out results for genes and clinical variables that were non-signicant. In terms of epigenetics and boosting the clinical value of this research I would further recommend, if possible NGS applications to delve into details of transcriptomics to look at genes that may be differentially expressed to provide more clues on the epigenetic suppression/activation of genes relevant to neurodevelopment. I would further then try use these genes as a fingerprint of gene expression linked to methylation by assessing whole genome bisulfite sequencing or even just pyrosequencing to detect any CpG sites that could potentially be methylated. Overall, this is great research with invaluable potential to contribute to both epigenetic regulation in the field of neuerbiology.
Author Response

(The authors gave the same response as above.)

Reviewer 3 Report
The study was well designed and presented. It is suitable for publication in current form.
Author Response
Dear reviewer,
We thank you very much for having read our manuscript and have provided us with very useful comments.
Best Regards,
Carla Lintas